# Metamaterial with Tunable Positive and Negative Hygrothermal Expansion Inspired by a Four-Fold Symmetrical Islamic Motif

Teik-Cheng Lim

School of Science and Technology, Singapore University of Social Sciences, Singapore 599494, Singapore; tclim@suss.edu.sg

**Abstract:** A metamaterial with controllable positive and negative thermal and hygroscopic expansions is investigated herein by inspiration from a range of Islamic geometric patterns. Constructing from eight pairs of pin-jointed Y-elements, each unit cell manifests eight rhombi that are arranged circumferentially, thereby manifesting four axes of symmetry. By attachment of bimaterial spiral springs of contrasting expansion coefficients to the far arms of the paired Y-elements, a change in the environment's thermal or hygroscopic condition alters the offset angle of the paired Y-elements such that the unit cell of the metamaterial ranges from the eight-pointed star to the regular octagon. The effective coefficient of thermal expansion (CTE) and the coefficient of moisture expansion (CME) of this metamaterial were developed for small and large changes in environmental fluctuations using infinitesimal and finite models, respectively. Generated data indicates that the sign and magnitude of the effective thermal and hygroscopic expansion coefficients can be controlled by geometrical descriptors of the bimaterial spiral spring—such as its coil number and the ratio of its mean radius to its thickness—as well as the properties of the bimaterial's layers such as their expansion coefficients, Young's moduli and, in the case of effective hygroscopic expansion, their relative absorptivity. The obtained results suggest that the proposed metamaterial can be designed to perform as highly sensitive thermal and/or moisture sensors, as well as other functional materials or devices that take advantage of environmental changes as stimuli.

**Keywords:** four-fold symmetry; Islamic geometrical patterns; metamaterial; negative moisture expansion; negative thermal expansion

## 1. Introduction

Interest in metamaterials may well be attributed to their capability of exhibiting novel, extreme and even counter-intuitive effective properties by precision design of their micro-lattice geometry. Examples of counter-intuitive effective mechanical properties have been extensively reviewed; these include auxetic (negative Poisson's ratio) [1–19], negative thermal expansion (NTE) [20–33], negative compressibility (NC) [34–40], negative moisture expansion (NME) [41–44] and combinations of two or more of these negative properties [45–53], to name a few.

Of late, metamaterials designs have been made possible by drawing inspiration from Islamic geometrical patterns [54–60]. A metamaterial that exhibits tunable positive and negative hygrothermal expansion is considered herein by inspiration from an Islamic geometrical pattern furnished in Figure 1, constructed by interlacing a regular octagon with four neighboring octagons along the on-axes directions and four smaller octagons in the diagonal directions. Such interlace produces a repetitive unit that exhibits a circumference of eight rhombi where the inner outline reveals an eight-pointed star, while reference to the smaller octagon shows a circumference of four rhombi in which a four-pointed star can be observed at its inner outline. The circumference of eight rhombi—exhibiting outer

and inner outlines of an octagon and an eight-pointed star, respectively—is a relatively common Islamic geometrical pattern, and one that manifests a high degree of symmetry.

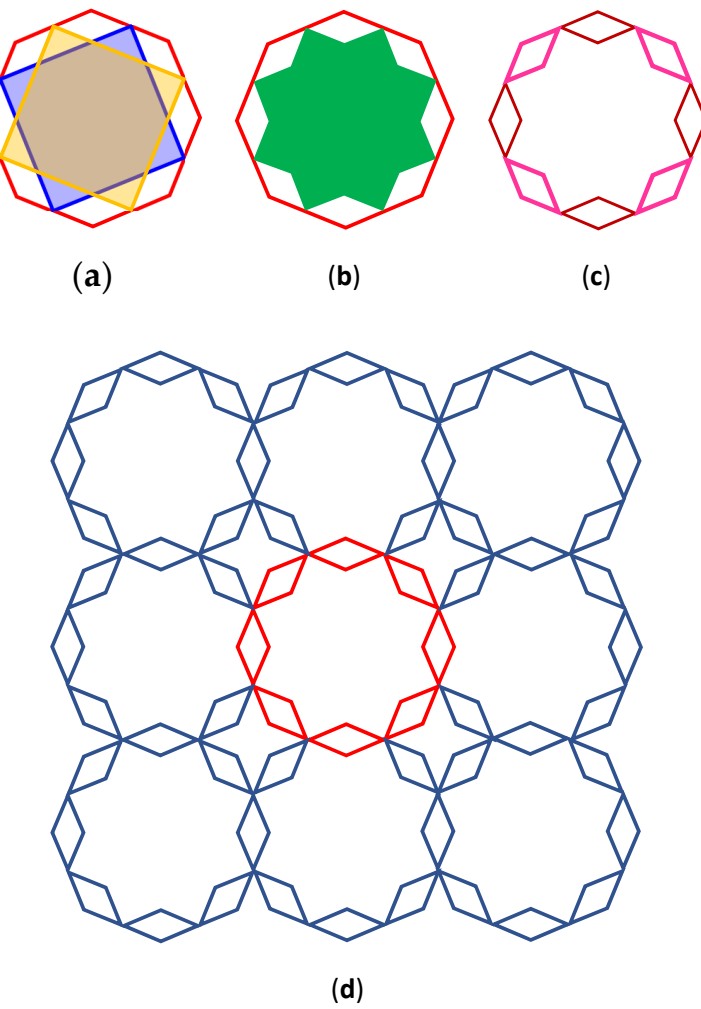

**Figure 1.** (**a**) A four-fold symmetrical unit cell of the metamaterial can be viewed as two overlapping squares offset by 45° from each other and enclosed by an octagon, such that (**b**) the combined outer outlines of both squares form an eight-pointed star known as the Rub-el-Hizb (ربع الحزب). (**c**) The voids between the octagon and the eight-pointed star can, in turn, be envisaged as a circular arrangement of eight rhombi, which constitutes a unit cell of the metamaterial. (**d**) A 3-by-3 array of the metamaterial units is interlaced by overlapping the rhombi aligned in the horizontal and vertical directions.

The metamaterial that is adopted herein is based on an assembly of paired counter-rotating rigid Y-elements. There has been precedence of paired counter-rotating rigid elements adopted for producing negative properties. For example, the Hoberman mechanism [61]—which is an assembly of paired linear elements pin-joined at the mid-span of the linear elements—has been employed for constructing negative thermal expansion (NTE) microstructures [62], and recently been shown to exhibit auxetic DNA structures [63,64]. If the linear element is replaced by equal-armed cross, i.e., an assembly consisting of paired counter-rotating crosses that are pin-joined at their centers, then one obtains a 2D auxetic system with Poisson's ratio is close to −1 [65], which was later extended to the 3D version [66]. In addition to auxetic properties, the counter-rotating crosses have been modified by attachment of ligaments of high coefficients of thermal expansion (CTE) and high coefficients of moisture expansion (CME) to give tunable positive to negative effective CTE and CME for the entire metamaterial [67]. An alternative modification to the paired

counter-rotating crosses that manifests Maltese crosses has also been shown to exhibit positive and negative hygrothermal expansion [68], as well as partially auxetic property [69]. Apart from the paired counter-rotating linear elements and paired counter-rotating crosses, paired counter-rotating Y-elements have been assembled to produce a 2D system with large isotropic negative thermal expansion [70]. A slight modification to the geometry of the Y-element permits the paired counter-rotating Y-elements to be assembled into an auxetic system that resembles the Islamic geometric pattern displayed in Figure 1 [71,72]. The ability of this highly symmetrical metamaterial to exhibit controllable positive and negative hygrothermal behavior is being explored in this paper. A comparison between previously designed metamaterials that are assemblies of pin-jointed counter-rotating rigid elements is furnished in Table 1, where the presence and absence of the corresponding negative properties are indicated as "Yes" and "No", respectively, while "NA" implies that the information is undefined, not applicable or unclear. Since the current metamaterial design replaces the uni-material spiral springs with bimetallic spiral springs, it follows that the negativity of Poisson's ratio discussed recently [71,72] applies to the current metamaterial. It can thus be seen from Table 1 that the currently considered metamaterial exhibits negativity in all three material properties—Poisson's ratio, coefficient of thermal expansion and coefficient of moisture expansion.

**Table 1.** Comparison between various paired counter-rotating rigid elements adopted for producing negative properties.

| Category | Shape of Paired Rigid or High Stiffness Elements | NPR | NTE | NME | Refs. |
|---|---|---|---|---|---|
| Hoberman mechanism | Rigid linear elements | Yes | No | No | [61] |
| Hoberman mechanism | Rigid linear elements with high CTE radial beams | NA | Yes | NA | [62] |
| Hoberman mechanism | High stiffness of approximately linear elements in some DNA | Yes | No | No | [63,64] |
| Equal armed crosses | Rigid equal armed cross elements | Yes | No | No | [65,66] |
| Equal armed crosses | Rigid equal armed cross elements with high CTE connecting beams | NA | Yes | Yes | [67] |
| Maltese crosses | Rigid equal armed cross elements with bimetallic spiral springs, hinge rods and connecting rods | NA | Yes | Yes | [68] |
| Maltese crosses | Rigid equal armed cross elements with spiral springs, hinge rods and connecting rods | Yes | No | No | [69] |
| Y-elements | Rigid Y-elements of equal angular spacing $(120°, 120°, 120°)$ and high CTE connectors | NA | Yes | NA | [70] |
| Y-elements | Rigid Y-elements of unequal angular spacing $(90°, 135°, 135°)$ with spiral springs | Yes | No | No | [71,72] |
| Y-elements | Rigid Y-elements of unequal angular spacing $(90°, 135°, 135°)$ with bimetallic spiral springs | Yes | Yes | Yes | This paper |

NPR = Negative Poisson Ratio, NTE = Negative Thermal Expansion, NME = Negative Moisture Expansion.

## 2. Analysis

The aim of this study is to establish the effective CTE and CME of the considered metamaterial for small and large environmental fluctuations using infinitesimal and finite deformation models, respectively. The design and setting of the study is described as follows. The geometry of each rigid Y-element, as shown in Figure 2a, consists of two near arms and one far arm, whereby the two near arms are aligned 90° from each other, while the near and far arms are oriented at 135° from each other. The length of all arms is equal, and is assigned the symbol $l$. The pair of Y-elements is offset by an angle $2\theta$, and connected at their junctions by a pin-joint as indicated in Figure 2b, whereby elastic restraint between the two Y-elements is implemented by a spiral spring that connects to the far arms of both Y-elements. The offset angle is defined as $2\theta$ so that its half-angle $\theta$ is used in the analysis. The effective Young's modulus arising from the spring of stiffness $k$ has been earlier

calculated for the case of very large number of unit cells [71], followed by investigation on the size effect and aspect ratio [72]. In a departure from previous studies, the spiral spring considered herein is made from bimaterial layers, as illustrated in Figure 2c, such that a change in environmental temperature and/or moisture concentration would alter the overall curvature of the spring. A total of eight pairs of Y-elements are assembled to form one unit cell in isolation, as shown in Figure 2d. It should be mentioned at this juncture that the number of Y-element pairs for each unit cell in the metamaterial under consideration is less than eight. This is because every pair of Y-element is shared by two adjacent unit cells such that every unit cell that is located in the interior of the metamaterial's boundary possesses an equivalent of four pairs of Y-elements. The number of paired Y-elements associated to a unit cell is five for those at the straight boundary and six for those at the corners.

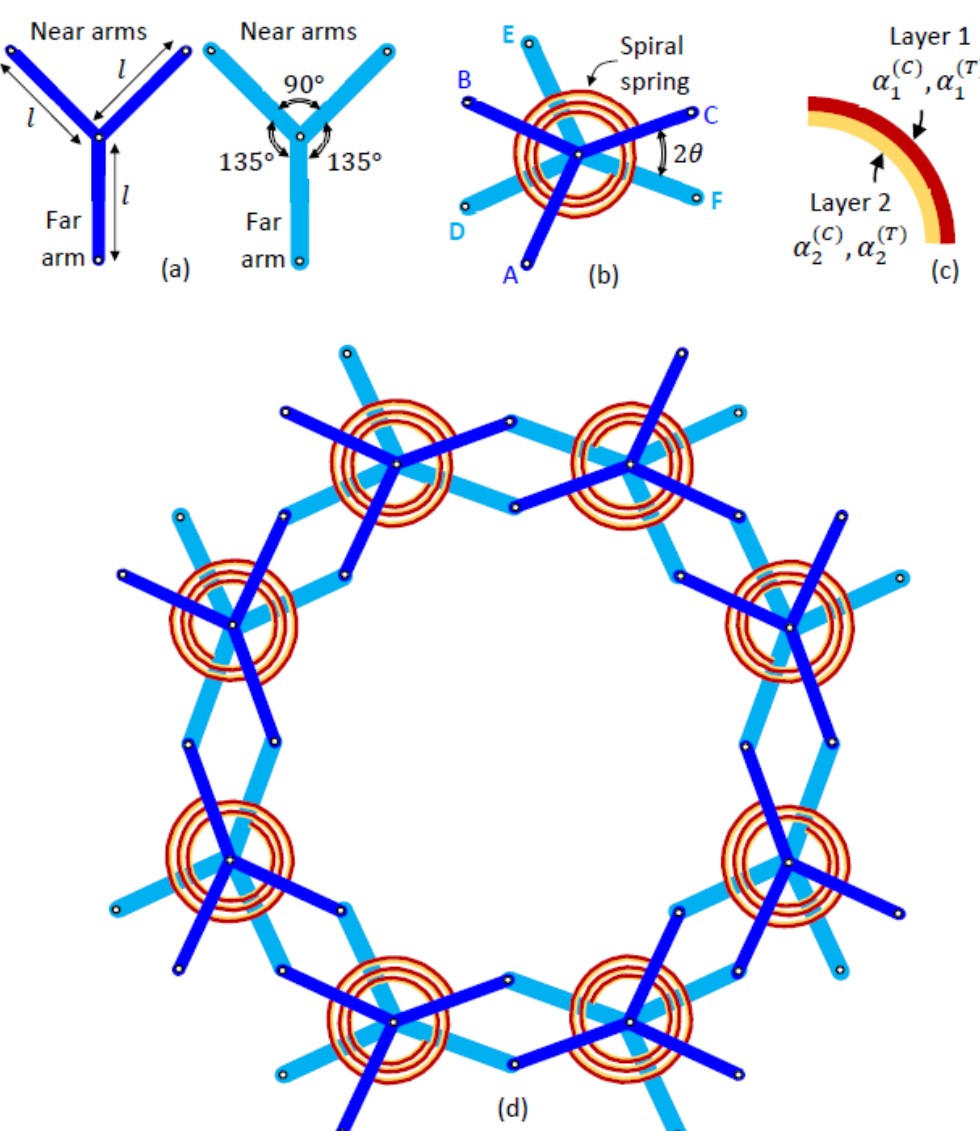

**Figure 2.** (**a**) Two Y-elements with their geometrical descriptions, (**b**) a pair of Y-elements connected at their junctions and elastically restrained by bimaterial spiral spring, showing their angular offset $2\theta$, (**c**) layers 1 and 2 of the bimaterial spring with contrasting CTEs and CMEs and (**d**) an assembly of eight pairs of Y-elements to form an isolated unit cell that resembles the four-fold symmetrical Islamic geometrical pattern furnished in Figure 1.

Arising from the temperature and/or moisture concentration fluctuation in the surrounding environment, the resulting change in the spring curvature alters the offset angle between the paired Y-elements such that the unit cell is re-shaped to give an updated size, which translates into an effective CTE and/or CME of the metamaterial. The manner in which the pair of Y-elements counter rotates, i.e., the change in angular offset influencing the unit cell assemblage—and hence its overall size—is displayed in Figure 3. Starting from the original state denoted by Figure 3 (top), the counter rotation of the paired Y-elements according to the unit contraction indicated in Figure 3 (left) causes the unit cell to undergo contraction until attaining the fully closed configuration, which resembles a typical Islamic eight-pointed star consisting of a circular arrangement of eight squares. Starting again from the original state, the counter rotation of the paired Y-elements according to the unit expansion denoted in Figure 3 (right) causes the unit cell to undergo expansion until attaining the fully opened configuration, which resembles a common motif of octagon and tilted square array. In other words, the deformation of the spiral spring as the environment changes can drive the change of geometrical parameters of metamaterial. In the present study, we consider the condition of the pin joints, or any other parts that are in contact but encounter relative motion, to be well-lubricated such that the effect of friction can be neglected.

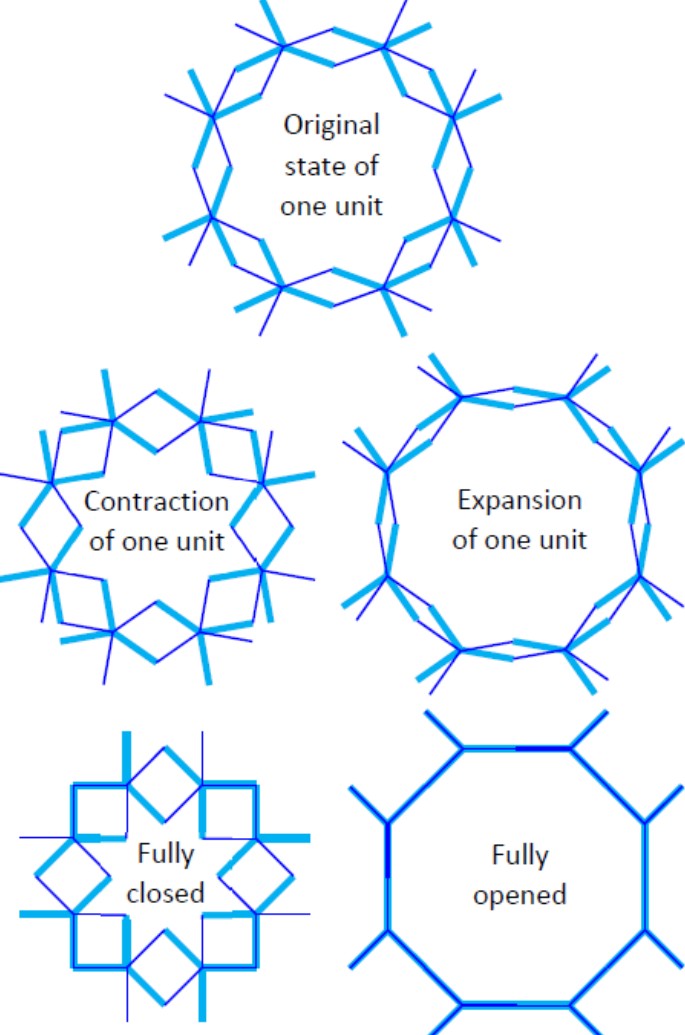

**Figure 3.** The original state of a unit cell (**top**) undergoing contraction (**left**) and expansion (**right**) until the attainment of fully closed and fully opened configurations, respectively (**bottom**).

The manner in which each unit cell is connected to its nearest neighbor is furnished in Figure 4. Specifically, Figure 4 (top) shows a 3-by-3 square array of unit cells in which $2\theta = 45°$. in the original state. The metamaterial encounters contraction when the paired Y-elements rotate in such a way that increases the offset angle as indicated in Figure 4 (left), and experiences expansion when the paired Y-elements rotate such that the offset angle decreases as denoted in Figure 4 (right) until their corresponding limiting configurations are attained (Figure 4, bottom). The range of offset angle is thus $45° < 2\theta < 90°$ for contraction and $0° < 2\theta < 45°$ for expansion, with $2\theta = 90°$ and $2\theta = 0°$ denoting the fully closed and fully opened configurations, respectively. Before proceeding with the analysis, it is imperative to lay down a list of assumptions that is adopted herein. It is assumed that (a) the Y-elements are rigid and possess zero CTE and zero CME such that they neither distort due to applied load nor encounter any change in size arising from temperature and/or moisture concentration fluctuations in the surrounding environment, (b) the difference in radial distance of both ends of the spiral spring is insignificant such that a mean radius of curvature of the spiral spring is representative for the entire spiral spring, (c) the metamaterial in its unconstrained state has no internal stress apart from that generated at the interface of the bimaterial strip and (d) the arc length of the bimaterial strip is conserved. The validity of the last assumption has been proven in the Appendix of Ref. [73].

The radius of the spiral spring is defined as the distance from center of the spiral spring to the mid-plane of the bimaterial strip. If both layers of the bimaterial strip possess equal thickness, then the radius of curvature is the distance from the center of the spiral spring to the bimaterial's interface. Since the radius increases monotonically from a minimum $r_{min}$ at the inner end to a maximum $r_{max}$ at the outer end, it is useful to quantify the mean radius of curvature as

$$r_m = \frac{1}{\Theta_{max}} \int_0^{\Theta_{max}} r \, d\Theta \tag{1}$$

where $\Theta$ is the angle swept from the inner end ($\Theta = 0$) to the outer end ($\Theta = \Theta_{max}$) of the spiral spring. The use of the mean spiral spring radius is valid when the difference between the maximum and minimum radii is small in comparison to the mean radius, $r_{max} - r_{min} \ll r_m$, as stated earlier in assumption (b). Consider a spiral spring of $n = 1$ number of coils, i.e., $\Theta_{max} = 2\pi$ as shown in Figure 5 (left) for the sake of simplicity, as it will later be shown that the coil number is never an integer except when $\theta = 0$. Suppose there is a change in environmental temperature and/or moisture concentration such that the mean curvature of the spiral spring changes. If the mean curvature decreases, i.e., the mean radius of curvature increases as shown in Figure 5 (middle), then a gap of $\delta\Theta$ is generated. If, on the other hand, the mean curvature increases, i.e., the mean radius of curvature decreases as indicated in Figure 5 (right), then an overlap of $\delta\Theta$ is produced. To prevent ambiguity, the resulting angular gap and angular overlap are assigned negative and positive values, respectively. If, instead of a single coil, the number of coils is generalized as $n$, then the total angular gap or angular overlap is correspondingly $n\delta\Theta$. Invoking the conservation of arc length, as indicated in assumption (d), the arc length based on the original mean radius of curvature $r_m(2\pi n)$ is equated to the arc length for the updated mean radius of curvature $r_m'(2\pi n + n\delta\Theta)$ to yield

$$r_m' = \frac{2\pi}{2\pi + \delta\Theta} r_m \tag{2}$$

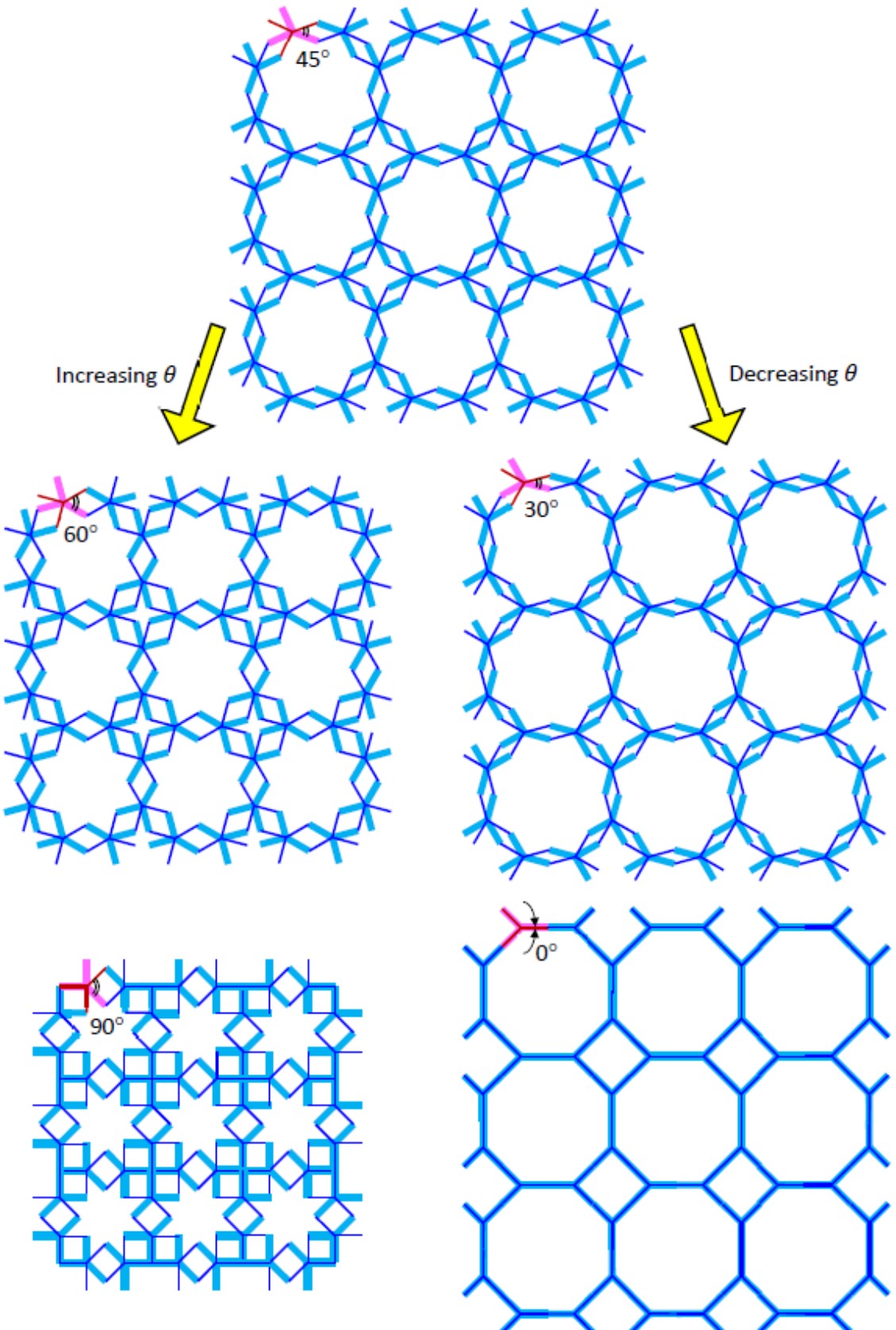

**Figure 4.** An interconnected 3-by-3 array of unit cells in its original state (**top**) undergoing contraction (**left**) and expansion (**right**) until attaining their limiting states (**bottom**).

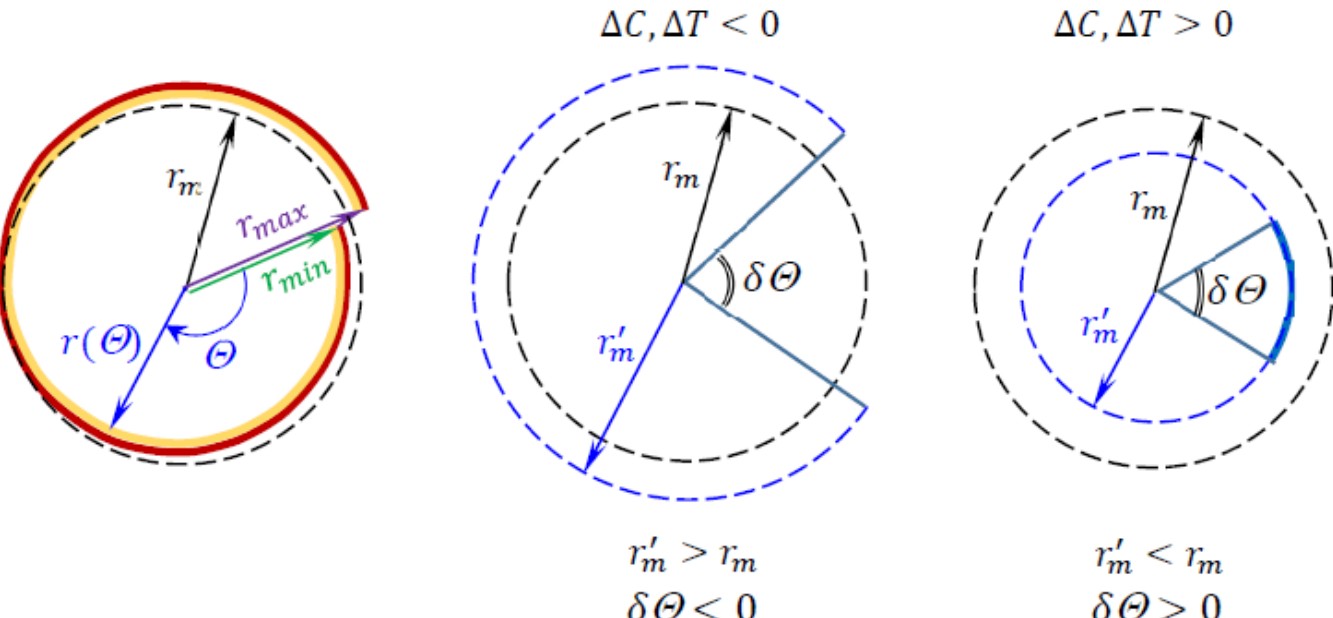

**Figure 5.** Visual representation of mean radius $r_m$ for a spiral spring of the variable radius $r = r(\Theta)$ for $n = 1$ number of coils, i.e., $0 < \Theta \leq 2\pi$ (**left**) for convenience, with increased (**middle**) and decreased (**right**) mean radius of curvature.

Let each Y-element rotate by $\Delta\theta$, then the relative change in offset angle between a pair counter-rotating Y-elements is $2\Delta\theta$, where $\Delta\theta < 0$ and $\Delta\theta > 0$ correspond to the metamaterial's overall expansion and contraction, respectively. Assigning CTEs of $\alpha_1^{(T)}$ and $\alpha_2^{(T)}$, CMEs of $\alpha_1^{(C)}$ and $\alpha_2^{(C)}$, Young's moduli of $E_1$ and $E_2$, and thicknesses of $h_1$ and $h_2$ to layers 1 and 2, respectively, of the bimaterial spring, the change in its mean curvature of the due to changes in the temperature $\Delta T$ and moisture concentration $\Delta C$ can be inferred from Timoshenko's model [74] for an initially curved bimetallic strip to yield

$$\frac{1}{r_m{'}} - \frac{1}{r_m} = \frac{\left(\alpha_1^{(T)} - \alpha_2^{(T)}\right)\Delta T}{\frac{h}{2} + \frac{2}{h}(E_1 I_1 + E_2 I_2)\left(\frac{1}{E_1 h_1} + \frac{1}{E_2 h_2}\right)} \tag{3}$$

and

$$\frac{1}{r_m{'}} - \frac{1}{r_m} = \frac{\alpha_1^{(C)}\Delta C_1 - \alpha_2^{(C)}\Delta C_2}{\frac{h}{2} + \frac{2}{h}(E_1 I_1 + E_2 I_2)\left(\frac{1}{E_1 h_1} + \frac{1}{E_2 h_2}\right)} \tag{4}$$

respectively, where $h = h_1 + h_2$ and the second moment area for each layer is

$$I_i = \frac{b h_i^3}{12} \tag{5}$$

for $i = 1, 2$, in which unit width ($b = 1$) for the bimetallic strip is prescribed. It is easily seen that if the bimaterial strip is initially straight, the usual model for the resulting radius of curvature is recovered by substituting $1/r_m = 0$. Based on the definition of moisture concentration $C = 100\, m/M$ where $m$ is the mass of water in a solid of dry mass $M$, one obtains the change in moisture concentration in layer $i$ of the bimaterial spring

$$\Delta C_i = 100\frac{\Delta m_i}{M_i} \tag{6}$$

where $\Delta m_i$ is the change in water mass in layer $i$, which is of dry mass $M_i$. For consistency, the change in environmental moisture concentration is similarly defined as

$\Delta C = 100 \Delta m_{env} / M_{env}$ for a change in moisture mass $\Delta m_{env}$ within an enclosure of air with dry mass $M_{env}$. Since the ends of the spiral spring are connected to the far arms of the paired Y-elements, the total change in angle formed by the ends of the spiral spring with its center $(n\delta\Theta)$ can be equated to the relative rotation of the paired Y-elements $(2\Delta\theta)$ to give

$$\delta\Theta = \frac{2\Delta\theta}{n} \tag{7}$$

Substituting Equations (2) and (7) into the LHS of either Equation (3) or Equation (4),

$$\frac{1}{r_m{'}} - \frac{1}{r_m} = \frac{1}{r_m}\frac{\delta\Theta}{2\pi} = \frac{1}{r_m}\frac{\Delta\theta}{\pi n} \tag{8}$$

Equating the RHS of Equations (3) and (4) with that of Equation (8) and considering the case where both layers of the bimaterial spring are of equal thickness $(h_1 = h_2 = h/2)$, leads to

$$\Delta\theta = \frac{24n\pi r_m}{h} \frac{\left(\alpha_1^{(T)} - \alpha_2^{(T)}\right)\Delta T}{14 + \frac{E_1}{E_2} + \frac{E_2}{E_1}} \tag{9}$$

where $\Delta T$ is the change in the temperature in layers 1 and 2 of the bimaterial spring at thermal equilibrium in response to a temperature change of $\Delta T$ in the environment, while

$$\Delta\theta = \frac{24n\pi r_m}{h} \frac{\alpha_1^{(C)}\Delta C_1 - \alpha_2^{(C)}\Delta C_2}{14 + \frac{E_1}{E_2} + \frac{E_2}{E_1}} \tag{10}$$

where $\Delta C_1$ and $\Delta C_2$ are the changes in the moisture concentration in layers 1 and 2 of the bimaterial spring, respectively, at hygroscopic equilibrium in response to a change in the moisture concentration $\Delta C$ in the environment.

Perusal to Equations (9) and (10) reveals that the rotation of each Y-element is significantly controlled by the number of coils $n$, the ratio of mean radius of curvature of the spiral spring to its thickness $r_m/h$, and the contrast between the expansion coefficients between the two layers of the bimaterial spring. As mentioned earlier, the coil number $n$ cannot be an integer due to the existence of an offset angle between the paired Y-elements. Since the angular offset $2\theta = 45°$ in the original state, the coil number can be either

$$n = \frac{2p - 1}{8} \tag{11}$$

or

$$n = \frac{2p + 1}{8} \tag{12}$$

where $p$ is an integer that must be chosen such that the coil number $p$ is about 2 or more practical reasons. Illustrations for the case where a substitution of $p = 12$ into Equations (11) and (12) are furnished in Figure 6 (left) and Figure 6 (right), respectively, with the bimaterial spring spiraling outward in clockwise and anti-clockwise directions shown in Figure 6 (top) and Figure 6 (bottom), respectively.

If the material properties are chosen such that $\alpha_1^{(T)} > \alpha_2^{(T)}$ and $\alpha_1^{(C)}dC_1 > \alpha_2^{(C)}dC_2$, then positive hygrothermal expansion (PHTE) is exhibited by Figure 6 (left), while negative hygrothermal expansion (NHTE) is manifested in Figure 6 (right).

Figure 7 displays the metamaterial's unit cell, with all the unnecessary lines removed for the sake of clarity, for establishing the infinitesimal models. The lines of length $l$ in Figure 7 correspond to those indicated in Figure 2a. In Figure 7, the lines in contact with the axes correspond to the far arms while those not in contact with the axes commensurate with the near arms, as compared with Figure 2d. The points Q and R refer to freely rotating pin joints.

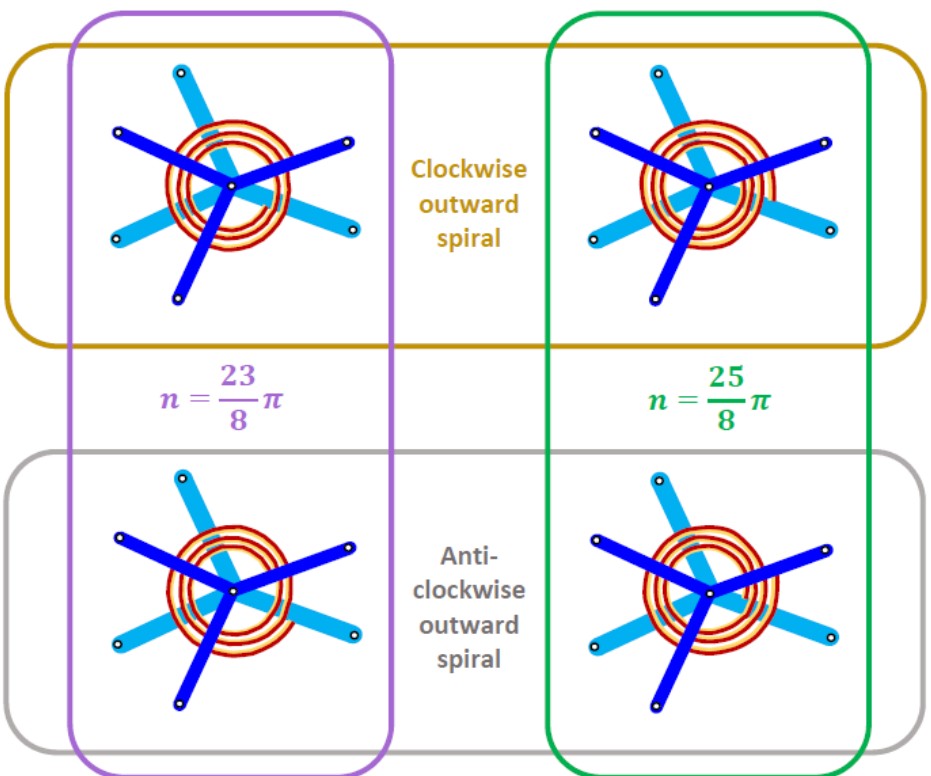

**Figure 6.** A pair of Y-elements where $\alpha_1^{(T)} > \alpha_2^{(T)}$ and $\alpha_1^{(C)} dC_1 > \alpha_2^{(C)} dC_2$ with clockwise outward spiral (**top row**) and anti-clockwise outward spiral (**bottom row**) of the bimaterial spring, such that $n = 23\pi/8$ (**left column**) and $n = 25\pi/8$ (**right column**).

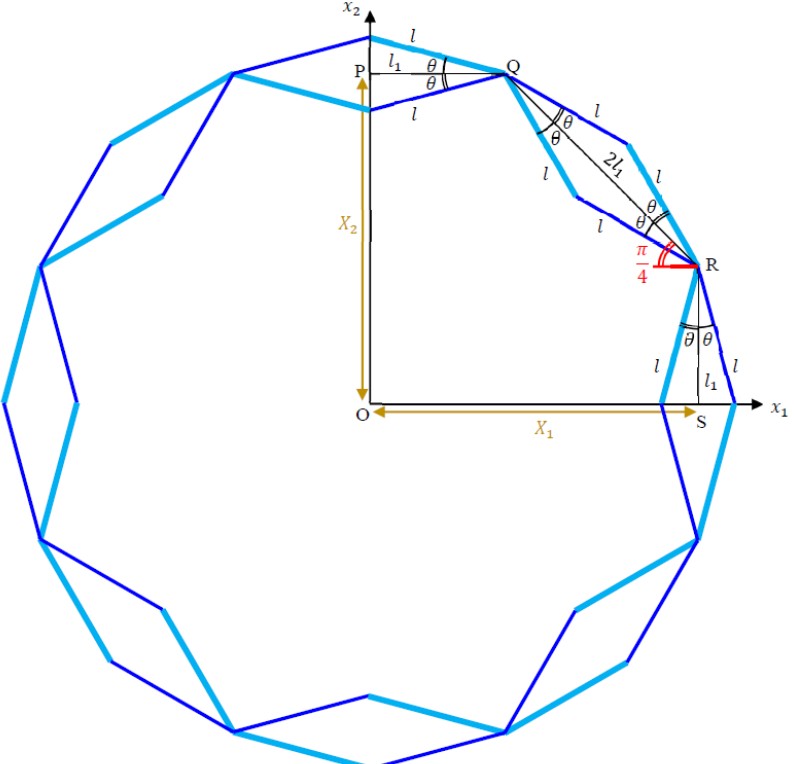

**Figure 7.** A unit cell with all the unnecessary lines removed for clarity. A quadrant of this unit cell is sufficient for analysis.

Taking the original distance P or S from the origin gives

$$X_i^0 = l_1 + 2l_1 \cos 45° = \left(1 + \sqrt{2}\right)l_1 \tag{13}$$

for $i = 1, 2$. Substituting $l_1 = l \cos\theta_0$ where

$$\cos\theta_0 = \frac{\sqrt{2 + \sqrt{2}}}{2} \tag{14}$$

for $\theta_0 = 22.5°$ in the original state leads to

$$X_i^0 = \left(1 + \sqrt{2}\right)l \cos\theta_0 = \left(1 + \frac{1}{\sqrt{2}}\right)^{\frac{3}{2}} l \tag{15}$$

With reference to Figure 7, the infinitesimal displacement of point P or point S along its corresponding axis is

$$dX_i^0 = \left(1 + \sqrt{2}\right)l(-\sin\theta_0 d\theta) \tag{16}$$

Substitution of

$$\sin\theta_0 = \frac{\sqrt{2 - \sqrt{2}}}{2} \tag{17}$$

for $\theta_0 = 22.5°$ in the original state leads to

$$dX_i^0 = -\sqrt{\frac{1}{2}\left(1 + \frac{1}{\sqrt{2}}\right)} \, l \, d\theta \tag{18}$$

The negative sign suggests that a decreasing $\theta$ gives an increasing $X_i$, or vice versa, which is consistent with the graphical representation furnished in Figure 4. The resulting nominal or engineering strain

$$\varepsilon_{ii}^0 = \frac{dX_i^0}{X_i^0} = -\frac{d\theta}{1 + \sqrt{2}} \tag{19}$$

would then lead to the following effective expansion coefficients

$$\alpha_{eff}^{(T)} = \frac{\varepsilon_{ii}^{(T)}}{dT} = -\frac{d\theta}{\left(1 + \sqrt{2}\right)dT} \tag{20}$$

$$\alpha_{eff}^{(C)} = \frac{\varepsilon_{ii}^{(C)}}{dC} = -\frac{d\theta}{\left(1 + \sqrt{2}\right)dC} \tag{21}$$

where $\varepsilon_{ii}^{(T)}$ and $\varepsilon_{ii}^{(C)}$ are the infinitesimal thermal and moisture strains in response to the environment's infinitesimal moisture concentration change $dC$ and temperature change $dT$, respectively. Substituting Equations (9) and (10) into Equations (20) and (21), one arrives at

$$\alpha_{eff}^{(T)} = -\frac{24n\pi r_m}{\left(1 + \sqrt{2}\right)h} \frac{\alpha_1^{(T)} - \alpha_2^{(T)}}{14 + \frac{E_1}{E_2} + \frac{E_2}{E_1}} \tag{22}$$

$$\alpha_{eff}^{(C)} = -\frac{24n\pi r_m}{\left(1 + \sqrt{2}\right)h} \frac{\alpha_1^{(C)}\frac{dC_1}{dC} - \alpha_2^{(C)}\frac{dC_2}{dC}}{14 + \frac{E_1}{E_2} + \frac{E_2}{E_1}} \tag{23}$$

The finite models are developed as follows. With reference to Equation (15) for which $0° < \theta < 45°$,

$$X_i = \left(1 + \sqrt{2}\right) l \cos \theta \tag{24}$$

The resulting logarithmic or true strain is thus

$$\varepsilon_{ii} = \ln \frac{X_i}{X_i^0} = \ln \frac{\cos \theta_0 \cos \Delta\theta - \sin \theta_0 \sin \Delta\theta}{\cos \theta_0} \tag{25}$$

Substituting

$$\tan \theta_0 = \sqrt{2} - 1 \tag{26}$$

from which $\theta_0 = 22.5°$ yields

$$\varepsilon_{ii} = \ln\left[\cos \Delta\theta - \left(\sqrt{2} - 1\right) \sin \Delta\theta\right] \tag{27}$$

Based on the definitions of thermal and moisture strain and substituting from Equations (9) and (10), one has

$$\alpha_{eff}^{(T)} = \frac{\varepsilon_{ii}^{(T)}}{\Delta T} = \frac{1}{\Delta T} \ln\left\{\cos\left[\frac{24n\pi r_m \left(\alpha_1^{(T)} - \alpha_2^{(T)}\right)\Delta T}{h(14 + E_1/E_2 + E_2/E_1)}\right]\right.$$
$$\left. - \left(\sqrt{2} - 1\right) \sin\left[\frac{24n\pi r_m \left(\alpha_1^{(T)} - \alpha_2^{(T)}\right)\Delta T}{h(14 + E_1/E_2 + E_2/E_1)}\right]\right\} \tag{28}$$

and

$$\alpha_{eff}^{(C)} = \frac{\varepsilon_{ii}^{(C)}}{\Delta C} = \frac{1}{\Delta C} \ln\left\{\cos\left[\frac{24n\pi r_m \left(\alpha_1^{(C)}\Delta C_1 - \alpha_2^{(C)}\Delta C_2\right)}{h(14 + E_1/E_2 + E_2/E_1)}\right]\right.$$
$$\left. - \left(\sqrt{2} - 1\right) \sin\left[\frac{24n\pi r_m \left(\alpha_1^{(C)}\Delta C_1 - \alpha_2^{(C)}\Delta C_2\right)}{h(14 + E_1/E_2 + E_2/E_1)}\right]\right\} \tag{29}$$

## 3. Results and Discussion

In evaluating the hygrothermal expansion behavior of the currently considered metamaterial, Equations (28) and (29) were used for plotting the effective CTEs $\alpha_{eff}^{(T)}$ versus environmental temperature change $\Delta T$ and the effective CMEs $\alpha_{eff}^{(C)}$ versus environmental moisture concentration change $\Delta C$, respectively. Since Equations (28) and (29) are undefined for $\Delta T = 0$ and $\Delta C = 0$, respectively, their infinitesimal counterparts as described by Equations (22) and (23) were used instead for calculating the effective CTE at $\Delta T = 0$ and the effective CME at $\Delta C = 0$. Apart from the expansion coefficients and Young's moduli, perusal to the infinitesimal and finite models of the effective expansion coefficients indicate that geometrical properties such as the coil number $n$, mean radius $r_m$ and thickness $h$ of the bimaterial spiral spring govern the overall expansion coefficients. The effect of spiral number will be evaluated with reference to Equations (11) and (12). Since the effects of mean radius and thickness of the bimaterial spiral spring are straightforward, i.e., $\alpha_{eff} \propto (r_m/h)$, a realistic value of $r_m/h = 200$ was set for calculating the effective expansion coefficients. In addition, the bimaterial spiral spring configuration was chosen such that it spirals clockwise outward, such as those illustrated in Figure 6 (top).

The effects of contrasting CTEs of the bimaterial layers $\left(\alpha_1^{(T)}, \alpha_2^{(T)}\right)$ and the coil number $n$ were selected for observing the effective CTE $\alpha_{eff}^{(T)}$ of the metamaterial with temperature change $\Delta T$. Table 2 lists the bimaterial systems selected for displaying the influence from contrasting CTEs of the layers. Figure 8 (left) shows the plots of the metamaterial's effective CTE versus $\Delta T$ for these three bimaterial systems for coil number $n = 25/8$ by substituting $m = 12$ into Equation (12). Since the copper-steel bimaterial is among the most widely used bimaterial strips, it was selected for observing the effect of coil number the

substitution of $m = 8$, 12, 16 into Equations (11) and (12) gives $n = 15/8$, 23/8, 31/8 and $n = 17/8$, 25/8, 33/8, respectively. Plots of the metamaterial's effective CTEs versus temperature change for these values of coil numbers are furnished in Figure 8 (right). The implication from these results is that the metamaterial can be used for developing highly sensitive and extremely precise thermally actuating materials or any other temperature-sensitive functional materials and devices. Perusal to Figure 8 shows three trends: (a) the difference in magnitude of the effective CTE for different bimetallic strip materials, (b) increased magnitude of the effective CTE for higher coil number $n$ and (c) increased magnitude of effective CTE for increased temperature with decreased magnitude of effective CTE for decreased temperature. The first trend (a) is attributed to the contrasting CTEs of the bimetallic layers—the larger the difference the greater is the change in curvature for the same change in temperature. The second trend (b) is a direct result of the coil number—for a higher number of coils, the change in subtending angle is greater for the same change in curvature of the bimetallic strip. The third trend (c) is due to geometrical consideration for the case whereby the CTE of the convex layer is higher than that of the concave layer $\alpha_1^{(T)} > \alpha_2^{(T)}$. Under this condition, an increased and decreased temperature of $|\Delta T|$ causes the curvature for the bimetallic strip to increase and decrease, respectively. Since the change in arc length is negligible, the corresponding decrease and increase in the radius of curvature produces a greater and smaller change in the subtending angle for $\Delta T > 0$ and $\Delta T < 0$, respectively, thereby resulting in a greater magnitude change in the effective CTE during heating and cooling, respectively.

**Table 2.** List of bimaterial layers' properties for plotting effective CTEs $\alpha_{eff}^{(T)}$.

| Bimaterial Spiral Spring System | CTE of Bimaterial Layers | Young's Modulus of Bimaterial Layers |
|---|---|---|
| (1) Tungsten (2) Silicon Carbide | $\alpha_1^{(T)} = 4.50 \times 10^{-6}$ K$^{-1}$ $\alpha_2^{(T)} = 2.77 \times 10^{-6}$ K$^{-1}$ | $E_1 = 405.0$ GPa $E_2 = 450.0$ GPa |
| (1) Carbon (2) Steel | $\alpha_1^{(T)} = 17.0 \times 10^{-6}$ K$^{-1}$ $\alpha_2^{(T)} = 12.0 \times 10^{-6}$ K$^{-1}$ | $E_1 = 117.0$ GPa $E_2 = 200.0$ GPa |
| (1) Brass (2) Titanium | $\alpha_1^{(T)} = 19.0 \times 10^{-6}$ K$^{-1}$ $\alpha_2^{(T)} = 8.60 \times 10^{-6}$ K$^{-1}$ | $E_1 = 112.5$ GPa $E_2 = 110.3$ GPa |

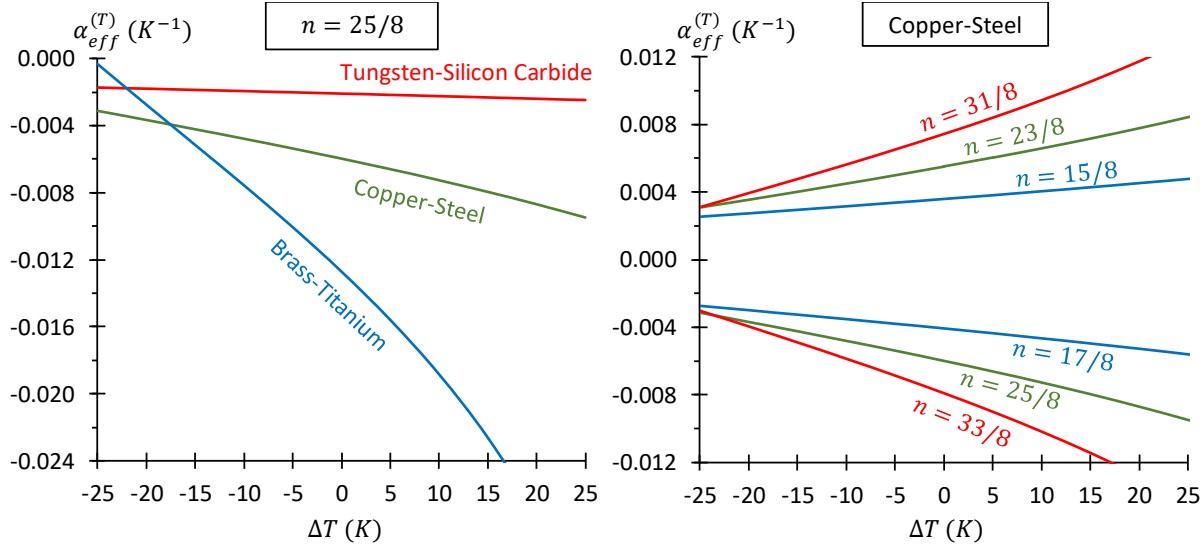

**Figure 8.** Plots of effective CTE versus temperature change for the tungsten-silicon carbide, copper-steel and brass-titanium bimaterial spiral spring at $n = 25/8$ (**left**), and for $m = 8$, 12, 16 (which gives two sets of coil number $n$) using copper-steel bimaterial spiral spring (**right**).

Comparison between models of the effective CTE $\alpha_{eff}^{(T)}$ with the effective CMEs $\alpha_{eff}^{(C)}$ indicates the obvious analogies between the bimaterial layers' individual CTEs $\left(\alpha_1^{(T)}, \alpha_2^{(T)}\right)$ with the bimaterial layers' individual CMEs $\left(\alpha_1^{(C)}, \alpha_2^{(C)}\right)$, as well as the analogy between the environment's temperature change $\Delta T$ and the environment's moisture concertation change $\Delta C$. In addition, influence from geometrical properties $(n, r_m/h)$ on the effective CTE is identical to the effective CME. As such, CMEs of the bimaterial layers and the coil number on the effective CME can be inferred from its thermal counterpart. Instead, an observation is made on the effect of the moisture concentration changes in the bimaterial layers $(\Delta C_1, \Delta C_2)$ relative to the environment's moisture-concentration change $\Delta C$. Due to the non-absorptivity of moisture in metallic solids, polymeric bimaterial spiral springs were selected for calculating the effective CMEs. Since the range of CME of typical polymers is typically $2 \times 10^{-3} \le \alpha^{(C)} \le 5 \times 10^{-3}$ [75–78], CMEs of $\alpha_1^{(C)} = 5 \times 10^{-3}$ and $\alpha_2^{(C)} = 2 \times 10^{-3}$ were selected. It should now be pointed out that if the absorptivity of both layers of the bimaterial spring are equal, leading to $\Delta C_1 = \Delta C_2$, then the effect of CME contrast $\left(\alpha_1^{(C)} - \alpha_2^{(C)}\right)$ on $\alpha_{eff}^{(C)}$ is comparable to the effect of CTE contrast $\left(\alpha_1^{(T)} - \alpha_2^{(T)}\right)$ on $\alpha_{eff}^{(T)}$. To observe only the effect of absorptivity, the present scope is limited to the case where $E_1 = E_2$ so as to simplify Equations (23) and (29) to

$$\alpha_{eff}^{(C)} = -\frac{3n\pi r_m}{2\left(1 + \sqrt{2}\right)h}\left(\alpha_1^{(C)}\frac{dC_1}{dC} - \alpha_2^{(C)}\frac{dC_2}{dC}\right) \tag{30}$$

and

$$\alpha_{eff}^{(C)} = \frac{1}{\Delta C}\ln\left\{\cos\left[\frac{3n\pi r_m\left(\alpha_1^{(C)}\Delta C_1 - \alpha_2^{(C)}\Delta C_2\right)}{2h}\right]\right.$$
$$\left. -\left(\sqrt{2} - 1\right)\sin\left[\frac{3n\pi r_m\left(\alpha_1^{(C)}\Delta C_1 - \alpha_2^{(C)}\Delta C_2\right)}{2h}\right]\right\} \tag{31}$$

respectively. Variation of $\Delta C_i/\Delta C$ (with $i = 1, 2$) was implemented via $\Delta C_1/\Delta C + \Delta C_2/\Delta C = 1$ so as to impose two extreme conditions whereby $\Delta C_1 = \Delta C$ when $\Delta C_2 = 0$ and $\Delta C_2 = \Delta C$ when $\Delta C_1 = 0$. Plots of the effective CMEs $\alpha_{eff}^{(C)}$ against the moisture concentration change in the environment $\Delta C$ are shown in Figure 9 for coil number $n = 23/8$ and $n = 25/8$ based on the outward spiral displayed by Figure 6 (top). It can be seen that the choice of $n = 23/8$ and $n = 25/8$ leads to mostly positive and negative $\alpha_{eff}^{(C)}$, respectively. When the absorptivity of layer 1 is very small compared to that of layer 2 of the bimaterial strip such that $\alpha_1^{(C)}\Delta C_1 < \alpha_2^{(C)}\Delta C_2$ (even though $\alpha_1^{(C)} > \alpha_2^{(C)}$), the signs of the effective CME reverse, i.e., negative and positive signs are observed for $n = 23/8$ and $n = 25/8$, respectively. The very high sensitivity towards moisture concentration change implies that this metamaterial can also be used for developing highly sensitive moisture-activating material or any other moisture-sensitive functional materials and devices. The trends of the effective CME in Figure 9 closely follow that of Figure 8 (with the moisture effects from $\alpha_1^{(C)}$, $\alpha_2^{(C)}$ and $\Delta C$ being analogous to the thermal effects from $\alpha_1^{(T)}$, $\alpha_2^{(T)}$ and $\Delta T$), except that the effective CME changes very gradually for $\Delta C_1 = 0$ and $\Delta C_2 = \Delta C$, but exhibits an exponential-like profile for $\Delta C_1 = \Delta C$ and $\Delta C_2 = 0$. The reason for this observation may well be attributed to the contrast between $\alpha_1^{(C)}\Delta C_1$ and $\alpha_2^{(C)}\Delta C_2$ in Equation (29) as opposed to the contrast of $\alpha_1^{(T)}\Delta T$ and $\alpha_2^{(T)}\Delta T$ in Equation (28). Since we have assigned $\alpha_1^{(C)} > \alpha_2^{(C)}$, it follows that the contrast is muted when $\Delta C_1 < \Delta C_2$, but the contrast becomes enhanced when $\Delta C_1 > \Delta C_2$.

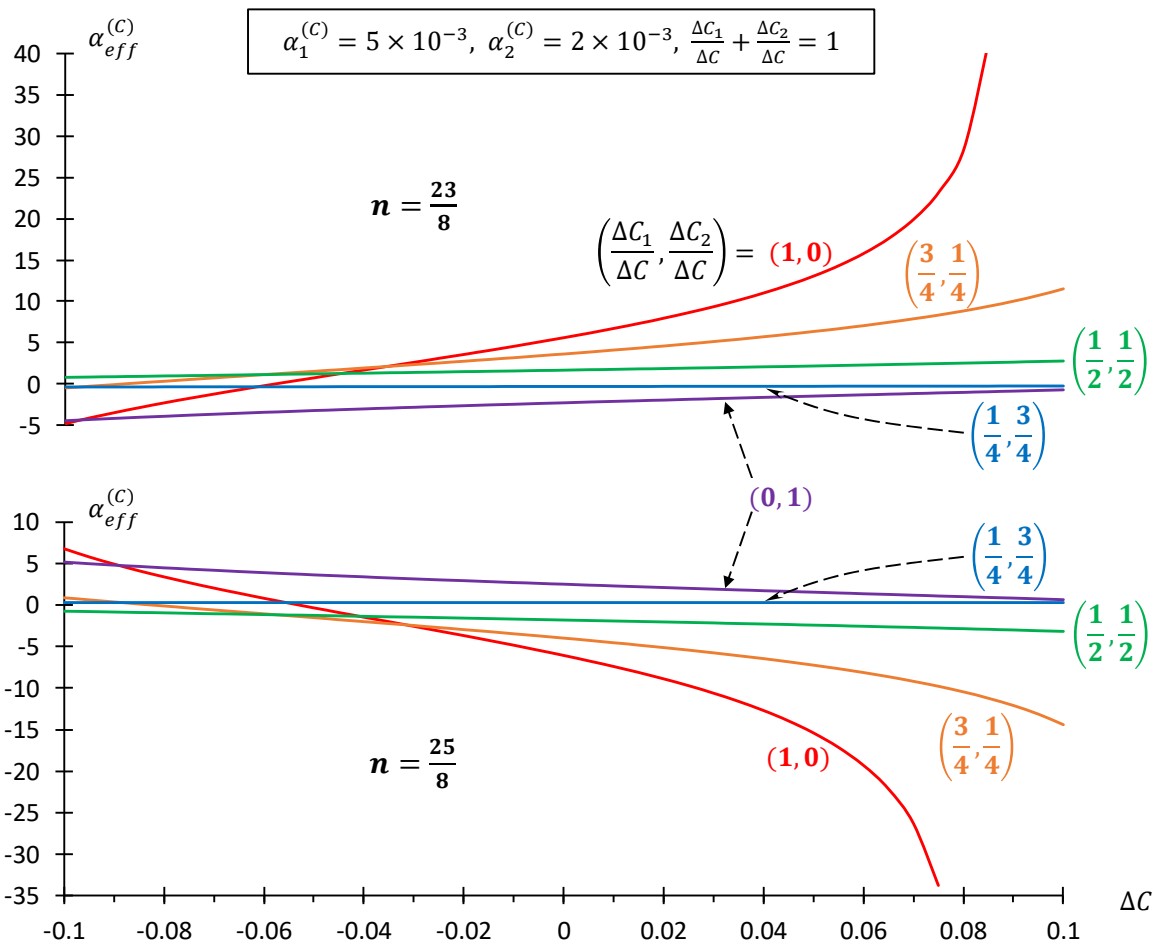

**Figure 9.** Plots of effective CME $\alpha_{eff}^{(C)}$ versus change in environmental moisture concentration $\Delta C$ for coil number $n = 23/8$ (**top**) and $n = 25/8$ (**bottom**) where $\alpha_1^{(C)} = 5 \times 10^{-3}$ and $\alpha_2^{(C)} = 2 \times 10^{-3}$ with the imposed condition $\Delta C_1 + \Delta C_2 = \Delta C$. An almost symmetrical profile between the family of $n = 23/8$ curves and the family of $n = 25/8$ curves can be observed.

It should now be mentioned that a limitation exists when the metamaterial is used as a moisture-concentration sensor. This is because some cases of small changes in moisture concentration, in which there is an accompanying large change in temperature. Even when the choice of bimaterial polymer layers have no appreciable difference in their layers' CTEs, the large temperature change coupled with a small moisture temperature change causes significant error to the moisture change readings due to the large temperature change. To counter such limitations, it is suggested that two sets of metamaterial sensors be used. In one set, the spiral springs are made from two layers of different metallic materials with contrasting CTEs, but are non-absorptive since they are metals, so that it responds solely due to temperature changes. The other set of metamaterial sensors uses spiral springs made from two layers of different polymeric materials with contrasting CMEs and insignificant, but inevitable, CTE contrast, which causes combined thermal and moisture-concentration changes. The sole effect from moisture-concentration change can therefore be accounted for by offsetting the temperature change readings from the combined thermal and moisture concentration changes.

## 4. Conclusions and Recommendation

A metamaterial with four-fold symmetry that was previously shown to possess perfect auxeticity via inspiration from a range of Islamic geometric patterns—by assembly from pairs of counter-rotating Y-elements [71,72]—has been modified herein to exhibit control-

lable positive to negative hygrothermal expansion behavior. The factors that influence the sign of the effective CTE and CME are (a) the positioning of the bimaterial's layers of higher and lower expansion coefficients—whether on the concave or convex side—and (b) the attachment of the bimaterial spring to the far arms of the paired Y-elements, which can spiral outwardly in clockwise or anti-clockwise directions. Point (b) would also influence the coil number, which is linearly proportional to the effective expansion coefficients. Other influencing parameters include the ratio of mean radius to thickness of the bimaterial spring $r_m/h$, the expansion coefficients of the bimaterial layers $\left(\alpha_1^{(T)}, \alpha_2^{(T)}, \alpha_1^{(C)}, \alpha_2^{(C)}\right)$ as well as the moisture absorptivity, which governs the changes to the moisture concentrations in the bimaterial layers $(\Delta C_1, \Delta C_2)$. Although the Young's modulus ratio of these bimaterial layers $(E_1/E_2)$ contributes to the effective expansion coefficients, its influence on the effective CTE is not substantial. Due to the ease of large deformation in this metamaterial in response to fluctuation in temperature and moisture concentration, it is suggested that the designed metamaterial can be tailor-made to perform as highly responsive sensors or other devices that take advantage of environmental changes as stimuli.

In view of the promising results from the current theoretical analysis, it is therefore suggested that finite element and/or experimental verification be performed so as to guide further refinement to the current models of expansion coefficients for this metamaterial. The refinement would include the consideration of friction at the pin joints or any other contacting movable parts in order to model the expansion coefficients under more realistic conditions.

**Funding:** This research received no external funding.

**Data Availability Statement:** Raw data of material properties is furnished in Table 2, which was used for calculating the effective CTE. Raw data for calculating the effective CMEs are based on references [75–78].

**Conflicts of Interest:** The author declares that he has no competing interests.

## Notations

| | |
|---|---|
| $\alpha_1^{(C)}, \alpha_2^{(C)}$ | Coefficients of moisture expansion for layers 1 and 2 of the bimaterial spiral spring. |
| $\alpha_1^{(T)}, \alpha_2^{(T)}$ | Coefficients of thermal expansion for layers 1 and 2 of the bimaterial spiral spring. |
| $\alpha_{eff}^{(C)}, \alpha_{eff}^{(T)}$ | Effective coefficients of moisture and thermal expansions of metamaterial. |
| $C$ | Moisture concentration. |
| $C_1, C_2$ | Moisture concentrations in layers 1 and 2 of the bimaterial spiral spring. |
| $\varepsilon$ | Strain |
| $\varepsilon_{ii}^{(C)}, \varepsilon_{ii}^{(T)}$ | Moisture and thermal strains in $Ox_i$ ($i = 1, 2$) directions. |
| $E_1, E_2$ | Young's moduli in layers 1 and 2 of the bimaterial spiral spring. |
| $h$ | Thickness of bimaterial spiral spring. |
| $h_1, h_2$ | Thicknesses of layers 1 and 2 of the bimaterial spiral spring. |
| $I_1, I_2$ | Second moment area for layers 1 and 2 of the bimaterial spiral spring. |
| $l$ | Arm length of Y-element. |
| $l_1$ | Half-distance between centers of two adjacent pairs of Y-elements. |
| $M$ | Dry mass. |
| $m$ | Moisture mass. |
| $n$ | Number of coils for the bimaterial spiral spring. |
| $p$ | Positive integer. |
| $r_m$ | Mean radius of curvature of bimaterial spiral spring. |
| $r_m{}'$ | Updated radius of curvature of bimaterial spiral spring. |
| $r_{max}$ | Maximum radius of curvature of bimaterial spiral spring. |
| $r_{min}$ | Minimum radius of curvature of bimaterial spiral spring. |
| $\theta$ | Half-angular offset for a paired Y-elements. |
| $\Theta$ | Angle formed between inner and outer ends of the spiral spring. |
| $T$ | Environmental temperature |
| $X_1, X_2$ | Half-dimensions of a unit cell measured along the $Ox_1$ and $Ox_2$ axes. |

## Abbreviations

CME     Coefficient of moisture expansion
CTE     Coefficient of thermal expansion
NC      Negative compressibility
NME     Negative moisture expansion
NTE     Negative thermal expansion

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
