# Peer review of "Metamaterial with Tunable Positive and Negative Hygrothermal Expansion Inspired by a Four-Fold Symmetrical Islamic Motif"

_symmetry, doi:10.3390/sym15020462_

Round 1

Reviewer 1 Report

The reviewer has gone through the full-length manuscript Title: "Metamaterial with tunable positive and negative hygrothermal expansion inspired by an Islamic motif". This manuscript studies the matamaterials with tunable thermal and hygroscopic expansions. There are many issues that should be addressed before it was considered for publication.

(1)     In light of existing published work, the paper is short of experimental data and finite element simulation study, and therefore does not verify the correctness of the theoretical results.

(2)   Second moment area is generally defined as I=bh3/12, the definition of I of metamaterials studied in this manuscript is different. Therefore, please justify the validity of using the equation in this work.

(3)   Please provide the possible reasons behind the trends observed in Fig. 8 and Fig. 9. The authors mainly presented the results of the theory, likewise the results were not discussed in-depth.

(4) Please indicate that the deforms of the spiral spring as the environment changes can drive the change of geometrical parameters of metamaterial. Whether you need to consider the effect of friction.

Author Response

The author is thankful the Reviewer #1, whose comments and suggestions help to improve the paper.

Reviewer 2 Report

Lim presents their findings on an Islamic motif-based metamaterial to observe positive and negative hygrothermal expansions. The study is analytically analyzed and it sounds scientific. I have a few comments to improve the quality of the paper.

-        First of all, the author must briefly mention the result of the study in the abstract and conclusion sections so that a layman can understand the author's contribution.

-        The previous studies are well analyzed by referring to similar geometrical approaches and similar purposes to this study. However, the author must be analyzed the previous studies by giving more specific reasons. The paper must contain a comparison table with them to understand the improvement.

-        In Fig 9, there is a nonlinearity with respect to the moisture concentration. How can the author explain this?  

Author Response

The author is thankful to Reviewer #2, whose comments and suggestions help to improve the paper.

Round 2

Reviewer 1 Report

The authors answered my questions and I recommend that the manuscript be accepted.